# Quantitative lung lesion features and temporal changes on chest CT in patients with common and severe SARS-CoV-2 pneumonia

Yue Zhang[1☯], Ying Liu[2☯], Honghan Gong[3], Lin Wu[3]*

1 Health Management Center, The First Affiliated Hospital, Nanchang University, Nanchang, Jiangxi Province, China, 2 Department of Emergency Medicine, The First Affiliated Hospital, Nanchang University, Nanchang, Jiangxi Province, China, 3 Department of Radiology, The First Affiliated Hospital, Nanchang University, Nanchang, Jiangxi Province, China

☯ These authors contributed equally to this work.
* linwuncu@163.com

**Data Availability Statement:** All relevant data are within the manuscript and its Supporting Information files.

## Abstract

The purpose of this study was to describe the temporal evolution of quantitative lung lesion features on chest computed tomography (CT) in patients with common and severe types of severe acute respiratory syndrome coronavirus 2 (SARS-CoV-2) pneumonia. Records of patients diagnosed with SARS-CoV-2 pneumonia were reviewed retrospectively from 24 January 2020 to 15 March 2020. Patients were classified into common and severe groups according to the diagnostic criteria of severe pneumonia. The quantitative CT features of lung lesions were automatically calculated using artificial intelligence algorithms, and the percentages of ground-glass opacity volume (PGV), consolidation volume (PCV) and total lesion volume (PTV) were determined in both lungs. PGV, PCV and PTV were analyzed based on the time from the onset of initial symptoms in the common and severe groups. In the common group, PTV increased slowly and peaked at approximately 12 days from the onset of the initial symptoms. In the severe group, PTV peaked at approximately 17 days. The severe pneumonia group exhibited increased PGV, PCV and PTV compared with the common group. These features started to appear in Stage 2 (4–7 days from onset of initial symptoms) and were observed in all subsequent stages ($p<0.05$). In severe SARS-CoV-2 pneumonia patients, PGV, PCV and PTV began to significantly increase in Stage 2 and decrease in Stage 5 (22–30 days). Compared with common SARS-CoV-2 pneumonia patients, the patients in the severe group exhibited increased PGV, PCV and PTV as well as a later peak time of lesion and recovery time.

## Introduction

Severe acute respiratory syndrome coronavirus 2 (SARS-CoV-2) pneumonia was prevalent in China from December 2019 to March 2020 and is currently being controlled effectively. However, SARS-CoV-2 cases are still rapidly increasing in other countries, representing a significant threat to global health [1–6]. SARS-CoV-2 is highly contagious and may lead to acute

**Funding:** This study was supported by the Natural Science Foundation of Jiangxi, China (Grant no. 2017BAB215048), the Science and Technology Project of Jiangxi Health Committee (Grant no. 20181020), and the Science and Technology Research Project of Jiangxi Provincial Department of Education (Grant no. 700993003).

**Competing interests:** The authors have declared that no competing interests exist.

respiratory distress or multiple organ failure [7, 8]. In SARS-CoV-2 pneumonia, chest computed tomography (CT) plays a key role in the diagnosis and evaluation of lesions [9, 10]. Previous studies described the shape, distribution and CT score of lesions and demonstrated diverse and rapidly changing chest CT manifestations of SARS-CoV-2 pneumonia [11–14]. These studies also reported ground-glass opacity (GGO) and consolidation lesions as two main features [15–18]. However, few studies on quantitative lung lesion features and temporal changes of SARS-CoV-2 pneumonia have been reported. Recently, artificial intelligence (AI) has demonstrated great success in the medical imaging domain due to its feature extraction ability, and this technique automatically classifies typical interstitial pneumonia using regional volumetric texture analysis in high-resolution CT [19, 20], representing a new paradigm for precisely describing the severity of SARS-CoV-2 pneumonia. Thus, we aimed to describe the evolution of quantitative lung lesion features in both severe and common SARS-CoV-2 pneumonia using an AI system with high-resolution CT imaging. We hope that our findings will help physicians objectively estimate temporal changes in the percentage of lung lesion volume in common and severe SARS-CoV-2 pneumonia patients.

## Materials and methods

The Ethics of Committees of the First Affiliated Hospital of Nanchang University approved this study. Informed consent was waived for this retrospective study. Data were collected and analyzed anonymously.

### Participants

In this study, records for patients diagnosed with SARS-CoV-2 pneumonia were reviewed retrospectively from 24 January 2020 to 15 March 2020. Patients were diagnosed according to the preliminary diagnosis and treatment protocols from the National Health Commission of the People's Republic of China [21]. Confirmed patients were eligible if they were admitted within 7 days from onset of initial symptoms and underwent an initial chest CT examination. The exclusion criteria were as follows: (1) patients with no obvious abnormal CT findings (mild type pneumonia), (2) patients who developed critical pneumonia because they were reviewed by a bedside chest X-ray rather than a chest CT. Finally, patients in this study were classified into the common and severe group according to the diagnostic criteria of severe pneumonia [21].

### CT protocol

Noncontrast chest CT scans were performed using a single inspiratory phase with commercial multi-detector CT scanners (Philips Brilliance iCT, Philips Medical Systems, the Netherlands). CT images were acquired at end inspiration. The scans were reconstructed as axial images with a slice thickness of 1–1.5 mm (iDose 4, Philips Medical Systems, the Netherlands). Image analysis was performed using Radiology Information System/Picture Archiving and Communication System.

### Chest CT evaluation

In this study, the Intelligent Evaluation System for Novel Coronavirus Pneumonia (version 6.5, Hangzhou YITU Healthcare Technology Co., Ltd.) was employed as the thin-section image analysis tool. The system combined the convolutional neural network and thresholding methods for segmentation of left and right lungs and detection of patchy shadows. Based on threshold CT values in the pneumonia lesions, quantitative CT features of pneumonia lesions

were automatically calculated using artificial intelligence algorithms, including the percentages of lesion volume with ranges of -700~-500 Hounsfield units (HU), -500~60 HU, and -700~60 HU, representing the percentages of ground-glass opacity volume (PGV), consolidation volume (PCV) and total pneumonia lesion volume (PTV) in both lungs (S1 Fig). Normal lungs were defined with range of -1000~-700 HU. The distribution of CT values in lungs was calculated to obtain a histogram. Based on this, the software outlined various lesions in each layer of the scanned images, then acquired the volume of lesions by calculating the pixels of each area outlined. Radiologists (WL and GHH) discerned and recorded PGV, PCV and PTV values.

### Statistical analysis

Statistical analyses were performed using IBM SPSS Statistics Software (version 21; IBM, New York, USA) and R software (version 3.0; Statistical Computing c/o Institute, Vienna, Austria) (http://www.r-project.org/). Quantitative data were presented as the mean ± standard deviation (minimum-maximum) or median (quartiles). The counting data were presented as the percentage of the total. PGV, PCV and PTV as a function of time were assessed and graphed by using the curve estimation module from package of "ggplot" of R software. Comparisons of nonpaired and paired quantitative data were evaluated using the Mann-Whitney U test and Wilcoxon test according to the normal distribution of data as assessed by the Shapiro-Wilk test. An exact $p$ value was used due to the insufficient sample size, and a $p$ value of $< 0.05$ was defined as statistically significant.

## Results

### Patient characteristics

A total of 73 patients were included in the study (Table 1), including 53 cases of common pneumonia and 20 cases of severe pneumonia. The average age was 45 ± 14 years for patients with common pneumonia and 50 ± 15 years for patients with severe pneumonia. The median time of the first pulmonary CT scan obtained from the onset of symptoms was 4 days (quartiles = 2, 6 days) for common pneumonia and 5 days (quartiles = 2.25, 6 days) in severe pneumonia. Half of the patients with severe pneumonia exhibit comorbidities. The most common comorbidities were hypertension (11.3%) and hepatitis B (13.2%) for common pneumonia and diabetes (30%) and hypertension (30%) for severe pneumonia. The most prevalent presenting symptoms were fever (88.7% in common pneumonia, 95% in severe pneumonia) and cough (45.3% in common pneumonia, 60% in severe pneumonia) (Table 1).

### Pulmonary CT evaluation

In common pneumonia, the PTV increased slowly, peaked approximately 12 days after the onset of initial symptoms with the peak percentage ranging from 2.5 to 5%, and then gradually decreased (Fig 1). In severe pneumonia patients, the PTV rapidly increased and peaked approximately 17 days after the onset of initial symptoms with the peak percentage ranging from 22 to 25% (Fig 1). The temporal trends of PGV and PCV were generally consistent with PTV in both groups.

Five stages were identified from the onset of initial symptoms: Stage 1 (0–3 days), Stage 2 (4–7 days), Stage 3 (8–14 days), Stage 4 (15–21 days), and Stage 5 (22–30 days). In Stage 1, PTV, PGV and PCV were not significantly different between the two types of pneumonia. In Stage 2, the severe group exhibited significantly increased PTV, PGV and PCV compared with the common group ($p = 0.001$ for PTV, $p = 0.001$ for PGV and $p = 0.001$ for PCV). The difference persisted through Stages 3, 4 and 5 ($p < 0.05$) (Table 2).

**Table 1. Characteristics of patients with common and severe SARS-CoV-2 pneumonia.**

|  | Common | Severe |
|---|---|---|
|  | (n = 53) | (n = 20) |
| **Patient demographics** |  |  |
| Age, years, mean ± SD (min, max) | 45 ± 14 (17, 75) | 50 ± 15 (21, 83) |
| Sex, No. (%) |  |  |
| Male | 31 (58.5) | 11 (55.0) |
| Female | 22 (41.5) | 9 (45.0) |
| Interval time of first CT scan from onset, days, Median (Quartiles) | 4 (2, 6) | 5 (2.25, 6) |
| **Comorbid conditions, No. (%)** |  |  |
| Any | 16 (32.2) | 10 (50.0) |
| Diabetes Mellitus | 5 (9.4) | 6 (30.0) |
| High Blood Pressure | 6 (11.3) | 6 (30.0) |
| Hepatitis B | 7 (13.2) | 4 (20.0) |
| Others | 2 (3.8) | 1 (5.0) |
| **Initial signs and symptoms, No. (%)** |  |  |
| Fever | 47 (88.7) | 19 (95.0) |
| Cough | 24 (45.3) | 12 (60.0) |
| Sputum production | 7 (13.2) | 4 (20) |
| Fatigue weakness | 8 (15.1) | 4 (20) |
| Myalgia | 3 (5.7) | 2 (10) |
| Sore throat | 6 (11.3) | 3 (15) |
| Headache | 10 (18.9) | 1 (5.0) |
| Chills | 9 (17.0) | 2 (10.0) |
| Diarrhea | 1 (1.9) | 1 (5.0) |
| Nausea | 1 (1.9) | 1 (5.0) |
| Vomit | 1 (1.9) | 1 (5.0) |

Quantitative data are presented as the mean ± standard deviation (minimum-maximum) or median (quartiles range). The counting data are presented as the percentage of the total. No., numbers; CT, computed tomography.

In common pneumonia, no significantly differences in the PTV, PGV and PCV were noted between Stages 1 and Stage 2, Stages 2 and 3, as well as Stages 3 and 4. However, the percentage of lesions in Stage 5 was reduced compared with that in Stage 4 ($p = 0.002$ for PTV, $p = 0.003$ for PGV and $p = 0.001$ for PCV) (Table 3). In severe pneumonia patients, PTV, PGV and PCV began to increase from Stage 2 to Stage 4 and decreased in Stage 5. In the severe group, Stage-2 patients exhibited increased PTV, PGV and PCV compared with Stage-1 patients ($p = 0.004$

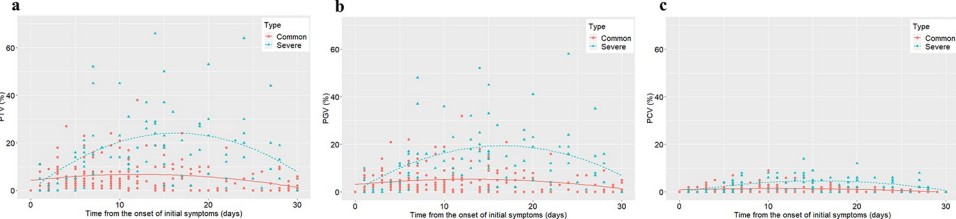

**Fig 1. Changes in PTV, PCV and PTV on chest CT from time of onset of initial symptoms (days).** Temporal changes in PTV (a), PGV (b) and PCV (c) for each patient. Fitted curve is depicted in each graph. PGV, percentage of ground-glass opacity volume; PCV, percentage of consolidation volume; PTV, percentage of total pneumonia lesion volume.

**Table 2. Comparison of PTV, PGV and PCV between common and severe SARS-CoV-2 pneumonia among five stages.**

| | Stage 1 (0–3 days) | | | Stage 2 (4–7 days) | | | Stage 3 (8–14 days) | | | Stage 4 (15–21 days) | | | Stage 5 (22–30 days) | | |
|---|---|---|---|---|---|---|---|---|---|---|---|---|---|---|---|
| | Common (n = 25) | Severe (n = 7) | p value | Common (n = 45) | Severe (n = 16) | p value | Common (n = 53) | Severe (n = 20) | p value | Common (n = 32) | Severe (n = 20) | p value | Common (n = 37) | Severe (n = 20) | p value |
| **Lesions, (%)** | | | | | | | | | | | | | | | |
| PTV | 4 (2, 8) | 2 (0, 5) | 0.223 | 3 (2, 7) | 14.5 (5.5, 20.25) | 0.001 | 4 (1, 9) | 21 (14, 29) | <0.001 | 5.5 (2, 10) | 20 (8, 29.5) | <0.001 | 1 (0, 4.5) | 12.5 (5, 20.75) | <0.001 |
| PGV | 3 (1, 7) | 1 (0, 5) | 0.205 | 2 (1, 5) | 11.5 (4.5, 16) | 0.001 | 3 (1, 6) | 16 (10, 22.75) | <0.001 | 5 (1.25, 8.75) | 15.5 (7, 25.25) | <0.001 | 1 (0, 3.5) | 10.5 (4, 18.25) | <0.001 |
| PCV | 1 (0, 2) | 0 (0, 1) | 0.273 | 1 (0, 1) | 3.5 (1, 4) | 0.001 | 1 (0, 2.5) | 5.5 (1, 8) | <0.001 | 1 (0, 2) | 3.5 (1.25, 5.75) | 0.001 | 0 (0, 0) | 2 (0, 5) | <0.001 |

Quantitative data were presented as median (quartiles range). A *p* value was calculated using the Mann-Whitney U test in this table. PGV, percentage of ground-glass opacity volume; PCV, percentage of consolidation volume; PTV, percentage of total pneumonia lesion volume.

for PTV, $p = 0.005$ for PGV and $p = 0.005$ for PCV), and Stage-3 patients exhibited slightly increased PTV, PGV and PCV compared with Stage-2 patients ($p = 0.048$ for PTV, $p = 0.036$ for PGV and $p = 0.056$ for PCV). PTV and PGV were not significantly different between Stages 3 and 4 ($p = 0.081$ for PTV and $p = 0.278$ for PGV), whereas PCV was reduced from Stages 3 to 4 ($p = 0.006$). PTV, PGV and PCV in Stage 5 were reduced compared with that in Stage 4 ($p < 0.001$ for PTV, $p = 0.001$ for PGV and $p = 0.001$ for PCV) (Table 3).

## Discussion

In this study, we provided reliable data of the temporal and lesion-quantified patterns for both types of SARS-CoV-2 pneumonia. Severe pneumonia exhibited greater PTV, PGV and PCV than common pneumonia, and these features appeared in Stage 2 (4–7 days from onset of initial symptoms) and remained in all subsequent stages. Severe pneumonia exhibited later peak and recovery time of PTV, PGV and PCV compared with common pneumonia.

**Table 3. Comparison of PTV, PGV and PCV between stages.**

| | Stage 1 vs. Stage 2 *p* value | Stage 2 vs. Stage 3 *p* value | Stage 3 vs. Stage 4 *p* value | Stage 4 vs. Stage 5 *p* value |
|---|---|---|---|---|
| **Common pneumonia** | | | | |
| PTV | 0.963 | 0.996 | 0.746 | 0.002 |
| PGV | 0.846 | 0.676 | 0.527 | 0.003 |
| PCV | 1 | 0.915 | 0.909 | 0.001 |
| **Severe pneumonia** | | | | |
| PTV | 0.004 | 0.048 | 0.081 | <0.001 |
| PGV | 0.005 | 0.036 | 0.278 | 0.001 |
| PCV | 0.005 | 0.056 | 0.006 | 0.001 |

Wilcoxon test was used in the comparisons between Stages 3 and 4 as well as between Stages 4 and 5 in patients with severe pneumonia. Mann-Whitney U test was used for other comparisons in this table. Please refer to Table 2 for quantitative data of variables of each group. PGV, percentage of ground-glass opacity volume; PCV, percentage of consolidation volume; PTV, percentage of total pneumonia lesion volume.

Our results showed greater PTV, PGV and PCV in patients with severe pneumonia than in those with common pneumonia, demonstrating the close relationship between the extent of lung lesion involvement and pneumonia severity. This finding is consistent with previous results demonstrating that patients with severe SARS-CoV-2 pneumonia were more likely to exhibit increased involvement of GGO and consolidation lesions compared with patients with common SARS-CoV-2 pneumonia [22]. In this study, quantitative lung lesion features in SARS-CoV-2 pneumonia might provide a more accurate and objective description of the involvement of lesions compared with previous studies with nonquantifiable patterns. Moreover, we also found increased PTV, PGV and PCV in severe pneumonia patients at Stage 2 (4–7 days from onset of initial symptoms) compared with common patients. As a result, we hypothesized that lung lesion involvement in Stage 2 but not Stage 1 might represent a valuable marker to predict the severity of SARS-CoV-2 pneumonia.

In the severe group, PCV declined significantly from Stage 3 to Stage 4, whereas PGV and PTV were not reduced. During this stage, the density (Hounsfield Unit) of consolidation decreased due to lesion absorption, which resulted in a portion of the consolidation lesion developing into a GGO lesion. Thus, we hypothesized this process might explain why PGV and PTV were not significantly reduced. Thus, in the period from Stage 3 to Stage 4, lesions are absorbed slightly.

The limitations of this study include the lack of chest CT scans in Stage-4 (n = 32) or Stage-5 (n = 37) common pneumonia patients compared with Stage-3 (n = 53) patients. The lack of chest CT scans resulted in inaccurate assessment of lesion changes from Stage 3 to Stage 4. It is possible that some Stage-1 to Stage-3 patients showed mild symptoms and a low percentage of lesions, so these patients did not receive close medical observation.

## Conclusions

In conclusion, increased PTV, PGV and PCV are noted in severe SARS-CoV-2 pneumonia patients beginning at Stage 2 (4–7 days after the onset of initial symptoms) compared with common SARS-CoV-2 pneumonia patients. Patients with severe pneumonia exhibited increased PTV, PGV and PCV as well as later peak time of lesion percentage and recovery time compared with patients with common SARS-CoV-2 pneumonia. Quantitative lung lesion features in SARS-CoV-2 pneumonia can objectively describe temporal changes in the percentage of pneumonia lesions.

## Supporting information

**S1 Fig. Schematic diagram of AI calculation method for PGV, PCV and PTV.**
(DOCX)

## Acknowledgments

The authors thank the medical faculty who participant in the treatment and care of patients infected by SARS-CoV-2.

## Author Contributions

**Conceptualization:** Honghan Gong, Lin Wu.

**Data curation:** Ying Liu.

**Methodology:** Lin Wu.

**Project administration:** Honghan Gong, Lin Wu.

**Writing – original draft:** Yue Zhang, Ying Liu.

**Writing – review & editing:** Lin Wu.

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
