## [Decision Letter · Decision Letter 0]

22 May 2020

PONE-D-20-10439

Quantitative lung lesions and temporal changes on chest CT in patients with common and severe SARS-CoV-2 pneumonia

PLOS ONE

Dear Dr. Wu,

Thank you for submitting your manuscript to PLOS ONE. After careful consideration, we feel that it has merit but does not fully meet PLOS ONE’s publication criteria as it currently stands. Therefore, we invite you to submit a revised version of the manuscript that addresses the points raised during the review process.

The article is potentially interesting but you need to address reviewers' concerns.

We look forward to receiving your revised manuscript.

Kind regards,

Raffaele Serra, M.D., Ph.D

Academic Editor

PLOS ONE

Journal Requirements:

"This study was supported by the Natural Science Foundation of  Jiangxi, China (Grant no. 2017BAB215048), the Science and Technology Project of Jiangxi Health Committee (Grant no. 20181020), the Science and Technology Research Project of Jiangxi Provincial Department of Education (Grant no. 700993003)."

Additional Editor Comments (if provided):

The manuscript is potentially interesting but reviewers' concerns should be addressed before we can reconsider the manuscript for publication.

Reviewers' comments:

Reviewer's Responses to Questions

**Comments to the Author**

1. Is the manuscript technically sound, and do the data support the conclusions?

Reviewer #1: Partly

2. Has the statistical analysis been performed appropriately and rigorously? 

Reviewer #1: No

3. Have the authors made all data underlying the findings in their manuscript fully available?

Reviewer #1: No

4. Is the manuscript presented in an intelligible fashion and written in standard English?

Reviewer #1: No

5. Review Comments to the Author

Reviewer #1: Manuscript PONE-D-20-10439

Title: Quantitative lung lesions and temporal changes on chest CT in patients with common and severe SARS-CoV-2 pneumonia

Overview: Retrospective study of chest CT of 73 patients with severe acute respiratory syndrome coronavirus 2 pneumonia in patients with non-severe and severe disease. The findings showed that the common type pneumonia resulted in peak CT findings in 12 days from onset of symptoms, while the severe type had greater CT findings that peaked at 17 days and delined later than for common type. Overall, the study is somewhat interesting, but focuses on ony one radiological finding generated by a single proprietary mysterious algorithm that others hence may not be able reproduce. Also the terminology and figures need clarification to make the study easier to understand.

.

Strong points:

a) Provides some potentially valuable information on the temporal CT changes of lung disease in the acute and initial subacute recovery phase of disease

b) Important clinical issue

Weak points

a) The percent lung volume percentage is a very rough term and does not account for anatomic location, shape, number of, or other features of the lesions.

b) Some odd statistical analysis results. Some p-values were surprisingly small. Also, there is a need to correct the level of significance for the large number of comparison tests.

c) Terms must be more precise and used consistently in the text. The current writing makes it very confusing.

d) Lack of information on HU thresholds for GGO versus consolidation versus normal, and use of a particular software that is not widely used and where the analytical algorithm is not provided makes the findings very difficult to generalize / impossible for others to use.

Specific points

1. For paired comparisons, there are quite a few of them and so there needs to be some correction of the level of significance to account for this large number of comparisons

2. Some of the p-values are much smaller than expected. For example, for severe disease, the total lung opacities for stage 3 was 21% (interquartile 14, 29) and for stage 4 was 20% (interquartile 8, 29.5), meaning that there was quite a big standard deviation and nearly identical median values. Further, the sample size n was only 20, yet the p-value comparison for stage 3 and 4 was a shockingly low 0.08, which is impossibly small for such data. Similarly the finding of consolidation was 5.5% with interquartile range 1, 8 for Stage 3 and was 3.5 with interquartile range 1.25, 5.75 for Stage 4 for severe disease, which with an n=20 should not yield a particularly significant p-value, but here is reported as p=0.003 which is surprisingly significant.

3. The manuscript needs to use precise terms that are defined clearly. Line 89, the “proportion of inflammatory volume” should probably be “percent lung volume involved by pathological opacity” or some other more specific term that includes the unit of measure. Oddly, the term “proportion of inflammatory volume” is never used again in the manuscript except fig 1. Instead the undefined ambiguous term “percentage of lung lesions” is used. What does “percentage of lung lesions”mean? Is it the percent of the diseased lung that is GGO? The total volume of the lung that is pathological in HU value? Please define.

4. The HU ranges for normal versus GGO versus consolidation should be described – was it the same for all patients? Was it normalized or adjusted in any way for each patient? How exactly does the software assign the "proportion of inflammatory volume"?

5. In the results, the term “percentage of lesions” is ambiguous and should be replaced by a more understandable term throughout the manuscript. Does it mean “lung volume percentage involved by pathological lung opacity” or is it just lung volume percentage involved by GGO or lung volume percentage involved by consolidation? The figures also use different terms. Fig 1 uses one term while Fig 2 uses the term “percentage of lung involvement” which is different than the terms used elsewhere in the text.

6. The Y-axis of the figures needs to be the same between the three graphs (total, GGO, and consolidation). Having different y-axis makes it hard to compare.

6. PLOS authors have the option to publish the peer review history of their article (what does this mean?). If published, this will include your full peer review and any attached files.

Reviewer #1: No

---

## [Author Response · Author response to Decision Letter 0]

9 Jul 2020

Dear reviewers，

Thank you very much for your guidance and advice. Here are the responses for your specific points. 

1. For paired comparisons, there are quite a few of them and so there needs to be some correction of the level of significance to account for this large number of comparisons

Re: Thank you for your comments. It is a very valuable advice. The question is also the limit in the study for not every patient included underwent CT scan in each stage. We have made a statistical analysis of all the data, and we use the exact p value to replace the p value in Mann-both Whitney U test and Wilcoxon test.

2. Some of the p-values are much smaller than expected. For example, for severe disease, the total lung opacities for stage 3 was 21% (interquartile 14, 29) and for stage 4 was 20% (interquartile 8, 29.5), meaning that there was quite a big standard deviation and nearly identical median values. Further, the sample size n was only 20, yet the p-value comparison for stage 3 and 4 was a shockingly low 0.08, which is impossibly small for such data. Similarly the finding of consolidation was 5.5% with interquartile range 1, 8 for Stage 3 and was 3.5 with interquartile range 1.25, 5.75 for Stage 4 for severe disease, which with an n=20 should not yield a particularly significant p-value, but here is reported as p=0.003 which is surprisingly significant.

Re: Thank you for your comments. These two sets of comparative data have been checked and analyzed repeatedly, and the p value in previous edition has been also replaced by the exact p value, but it has little change. We tried to analyze the comparisons by Mann-Whitney U test rather than paired comparisons, the p value was shockingly increased. Thus, we think that shockingly ow or particularly significant p value is because these two comparisons are paired tests. 

3. The manuscript needs to use precise terms that are defined clearly. Line 89, the “proportion of inflammatory volume” should probably be “percent lung volume involved by pathological opacity” or some other more specific term that includes the unit of measure. Oddly, the term “proportion of inflammatory volume” is never used again in the manuscript except fig 1. Instead the undefined ambiguous term “percentage of lung lesions” is used. What does “percentage of lung lesions”mean? Is it the percent of the diseased lung that is GGO? The total volume of the lung that is pathological in HU value? Please define.

Re: Thank you for your comments. We are very sorry to confuse you by the inconsistent and ambiguous terms in our manuscript on lesion percentage. In previous version, both “proportion of inflammatory volume” and “percentage of lung lesions” mean “percentage of total lesion volume”. We have substituted these inconsistent terms by “the percentages of ground-glass opacity volume (PGV)”, “the percentages of consolidation volume (PCV)”, and the percentages of the total lesions volume (PTV)” in both lungs. The percentage of total lesions volume was defined with ranges of -700~-500 Hounsfield units (HU), that is, the sum area of both PGV and PCV.

4. The HU ranges for normal versus GGO versus consolidation should be described – was it the same for all patients? Was it normalized or adjusted in any way for each patient? How exactly does the software assign the "proportion of inflammatory volume"?

Re: Firstly, we have added it in the Material and Methods that percentages of lesion volume with ranges of -700~-500 Hounsfield units (HU), -500~60 HU, and -700~60 HU corresponded to percentages of ground glass opacity volume (PGV), consolidation volume (PCV) and total lesion volume (PTV), where total lesion volume was defined as the sum area of both PGV and PCV. The normal lung was defined with range of -1000~-700 HU. Secondly, data calculation method was the same for all patients. All patients with SARS-CoV-2 pneumonia were performed with the same CT scanner and the same scanning parameters. The data were also processed with the same software analysis parameters to ensure the consistency of these data. Thirdly, the software assigned the "proportion of inflammatory volume" by outlining the lesions in each layer of the scanned images, then acquired the volume of the lesions by calculating the pixels of each area outlined. The process also has been added in the Material and Methods.

5. In the results, the term “percentage of lesions” is ambiguous and should be replaced by a more understandable term throughout the manuscript. Does it mean “lung volume percentage involved by pathological lung opacity” or is it just lung volume percentage involved by GGO or lung volume percentage involved by consolidation? The figures also use different terms. Fig 1 uses one term while Fig 2 uses the term “percentage of lung involvement” which is different than the terms used elsewhere in the text.

Re: We are very sorry to confuse you by the inconsistent and ambiguous terms in our manuscript on lesion percentage. In previous version, percentage of lesions represented different means according to the conjunctions, eg. percentage of lesions of GGO or percentage of lesions of consolidation. Sometimes we omitted the conjunctions so that some confusion was produced. Thus, we have substituted these inconsistent terms by “the percentages of ground-glass opacity volume (PGV)”, “the percentages of consolidation volume (PCV)”, and the percentages of the total lesions volume (PTV)” in both lungs. 

6. The Y-axis of the figures needs to be the same between the three graphs (total, GGO, and consolidation). Having different y-axis makes it hard to compare.

Re: Thank you for your comments. We have made the Y-axis of the figures the same between the three graphs (total, GGO, and consolidation). The modified figure has been uploaded.

If you have any questions, please let me know. Look forward to your response. 

Sincerely yours

---

## [Editor Report · Decision Letter 1]

16 Jul 2020

Quantitative lung lesion features and temporal changes on chest CT in patients with common and severe SARS-CoV-2 pneumonia

PONE-D-20-10439R1

Dear Dr. Wu,

We’re pleased to inform you that your manuscript has been judged scientifically suitable for publication and will be formally accepted for publication once it meets all outstanding technical requirements.

Kind regards,

Prof. Raffaele Serra, M.D., Ph.D

Academic Editor

PLOS ONE

Additional Editor Comments (optional):

amended manuscript is acceptable
---

## [Editor Report · Acceptance letter]

17 Jul 2020

PONE-D-20-10439R1 

Quantitative lung lesion features and temporal changes on chest CT in patients with common and severe SARS-CoV-2 pneumonia 

Dear Dr. Wu:

I'm pleased to inform you that your manuscript has been deemed suitable for publication in PLOS ONE. Congratulations! Your manuscript is now with our production department. 

Kind regards, 

on behalf of

Prof. Raffaele Serra 

Academic Editor

PLOS ONE